# Do Seropositive Wild Boars Pose a Risk for the Spread of African Swine Fever? Analysis of Field Data from Latvia and Lithuania

**DOI:** 10.3390/pathogens12050723

**Published:** 2023-05-17

**Authors:** Edvīns Oļševskis, Marius Masiulis, Mārtiņš Seržants, Kristīne Lamberga, Žanete Šteingolde, Laura Krivko, Svetlana Cvetkova, Jūratė Buitkuvienė, Simona Pilevičienė, Laura Zani, Nicolai Denzin, Klaus Depner

**Affiliations:** 1Food and Veterinary Service, LV-1050 Riga, Latvia; martins.serzants@pvd.gov.lv (M.S.); kristine.lamberga@pvd.gov.lv (K.L.); 2Institute of Food Safety, Animal Health and Environment “BIOR”, LV-1076 Riga, Latvia; zanete.steingolde@bior.lv (Ž.Š.); laura.krivko@bior.lv (L.K.); svetlana.cvetkova@bior.lv (S.C.); 3State Food and Veterinary Service, 07170 Vilnius, Lithuania; marius.masiulis@vmvt.lt; 4Veterinary Academy, Lithuanian University of Health Sciences, 44307 Kaunas, Lithuania; 5Faculty of Veterinary Medicine, Latvia University of Life Sciences and Technologies, LV-3004 Jelgava, Latvia; 6National Food and Veterinary Risk Assessment Institute, 08409 Vilnius, Lithuania; jurate.buitkuviene@nmvrvi.lt (J.B.); simona.pileviciene@nmvrvi.lt (S.P.); 7Niedersächsisches Landesamt für Verbraucherschutz und Lebensmittelsicherheit (LAVES), 26203 Wardenburg, Germany; laura.zani@laves.niedersachsen.de; 8Friedrich-Loeffler-Institut, 17493 Greifswald, Germany; nicolai.denzin@fli.de (N.D.); klaus.depner@fli.de (K.D.)

**Keywords:** virus carriers, survivors, epidemiological role of seropositive animals

## Abstract

In 2020, ASF occurred in wild boars throughout Latvia and Lithuania, and more than 21,500 animals were hunted and tested for the presence of the virus genome and antibodies in the framework of routine disease surveillance. The aim of our study was to re-examine hunted wild boars that tested positive for the antibodies and negative for the virus genome in the blood (n = 244) and to see if the virus genome can still be found in the bone marrow, as an indicator of virus persistence in the animal. Via this approach, we intended to answer the question of whether seropositive animals play a role in the spread of the disease. In total, 2 seropositive animals out of 244 were found to be positive for the ASF virus genome in the bone marrow. The results indicate that seropositive animals, which theoretically could also be virus shedders, can hardly be found in the field and thus do not play an epidemiological role regarding virus perpetuation, at least not in the wild boar populations we studied.

## 1. Introduction

When African swine fever (ASF) was introduced to the Baltic States and Poland in 2014, the epidemiological role of wild boars became evident and a new infection cycle was described: the wild boar–habitat cycle [1,2]. The disease became endemic in many wild boar populations and persisted for months and years. In the wake of these events, the question of whether seropositive animals that have survived the disease are epidemiologically relevant arose, as they can be considered virus carriers and thus play a role in further perpetuation of the ASF virus. Although ASF is associated with very high case fatality rates, a certain small proportion of infected animals recover from the infection and survive [3,4]. In this context, whether such survivors may act as carriers of the virus and may contribute to virus spread within an affected population was and still is speculated and discussed [5,6,7,8,9,10].

A large number of experimental studies have been conducted with domestic pigs to investigate the role of potential carriers and to explore their role in disease spread. In a detailed literature review [11], the authors evaluated the experimental studies and came to the conclusion that two types of “survivors” can be defined: (i) pigs that do not die but develop a persistent infection, characterized by periodic viremia and often but not always accompanied by some signs of subacute to chronic disease, and (ii) pigs that clear the infection independently of the virulence of the virus and do not excrete the virus beyond 30 to 40 days after infection. It was concluded that none of the categories of survivors can be considered as “healthy” carriers, i.e., pigs that show no signs but have the long-term ability to shed the virus and to transmit the disease to susceptible animals. Localized virus persistence in lymphoid tissues may occur to some extent in seropositive survivors, which in theory may cause infection after oral uptake. To what extent this is relevant under field conditions and for wild boars is still not clear. So far, there is also no clear or unequivocal evidence from field observations that seropositive wild boars could play an epidemiological role in spreading the virus [2,11].

According to the strategic approach to the management of African swine fever for the European Union, all wild boars hunted in ASF-affected areas are subjected to ASF routine testing [12]. Blood sample are tested via PCR and ELISA for the presence of the ASF virus genome and ASF antibodies, respectively. If the ASF virus genome or antibodies are detected, the wild boar is considered infected and the carcass is to be safely disposed of.

In 2020, ASF was present all over the countries Latvia and Lithuania. The pre-reproductive wild boar population was estimated at about 15,300 wild boars in Latvia and 13,500 wild boars in Lithuania, which is around 0.2 wild boars per km^2^. However, these estimates should be considered rough figures of the populations under which the disease has spread. In Latvia, a total of 191 wild boar carcasses were found and tested, of which 99 were PCR-positive (51.8%). In Lithuania, 82 carcasses out of 177 were tested PCR-positive (46%). The geographical location of ASF cases is showed in Figure 1. An overview of the testing results from the found carcasses, as well as from the hunted wild boars, is given in Table 1.

The aim of our study was to take a closer look at seropositive wild boars that were shot in Latvia and Lithuania in the framework of routine surveillance in 2020.

## 2. Materials and Methods

Our aim was to re-examine all wild boars that tested positive for the antibodies and negative for the virus genome in the blood and to see if the virus genome can still be found in the bone marrow, as an indicator of virus persistence in the animal. Using this approach, we intended to answer the question of whether seropositive animals could be virus-positive in the bone marrow at the same time, and if they play a role in the spread of the disease. We proceeded as follows: as soon as the initial laboratory results from the hunted wild boars became available, we re-examined the seropositive animals by collecting bone marrow from a long bone (humerus) for further virological examination. About 3 g of tissue or ligaments was extracted from the cancellous part of the bone with a bone chisel and was placed in 7 mL homogenization tubes with 2.5 mL PBS and 9–10 ceramic beads. The tubes were then placed in a homogenizer (Bead Ruptor 24, OMNI International, Kennesaw, GA, USA) and homogenized at 6000 rpm for 30 s in two cycles. For DNA extraction, the suspension was clarified via centrifugation for 2 min at 3000 rpm. The IndiSpin Pathogen Kit (QIAGEN^®^ GmbH for INDICAL BIOSCIENCE, Leipzig, Germany) for DNA manual extraction was used according to the manufacturer’s instructions. In total, we succeed in examining bone marrow samples from 244 seropositive wild boars, 139 from Latvia and 105 from Lithuania (Table 1). For each animal, the date of first sampling and location of hunting (GIS coordinates) were registered. The laboratory testing was performed at the National Reference Laboratories for ASF in Latvia (Institute of Food Safety, Animal Health and Environment BIOR) and Lithuania (National Food and Veterinary Risk Assessment Institute (NMVRVI). For ASF virus/genome detection, the real-time PCR described by Fernández-Pinero et al. (2013) was used [13]. Samples with Ct values <40 were considered positive. For ASF virus antibody detection, the commercially available Ingezim PPA Compac blocking ELISA kit (11.PPA.K.3, Ingenasa, Madrid, Spain) was used. All positive and doubtful samples were additionally tested with the immunoperoxidase test (IPT) for confirmation. The IPT was performed according to the European Union Reference Laboratory (EURL) Standard Operating Procedure (SOP) for ASF (SOP/CISA/ASF/IPT/1, 2021).

The results of laboratory examinations of hunted and found dead wild boars from the routine ASF surveillance in Latvia and Lithuania were used as background information to describe the epidemiological situation in 2020. As part of these routine surveillance, blood samples from all hunted wild boars were examined via PCR and ELISA. However, only tissue samples from animals found dead were examined via PCR. Statistical analyses were conducted in R [14] with the package binom [15]. The confidence intervals of the reported proportions were calculated using the Clopper–Pearson method [16]. A chi-square statistic with Yates correction was used to compare the proportions [17]. A *p*-value of 0.05 or lower was assumed to be indicative of a significant difference.

## 3. Results

The test results of hunted and found dead wild boars in Latvia and Lithuania from May to December 2020 are presented in Table 1. The prevalence of PCR-positive samples was significantly higher in wild boars found dead (around 50%) than in hunted animals (<1%). The confidence intervals (95%) did not overlap for individual countries or for the whole. The proportion of PCR-positive animals (seropositive or seronegative) among hunted wild boars was significantly higher in Latvia than in Lithuania with X^2^ (1, N = 21,575) = 8.9, *p* = 0.0029. On the contrary, the proportion of seropositive but PCR-negative animals (X^2^ (1, N = 21,575) = 3.4, *p* = 0.066) did not differ significantly between the two countries. The prevalence of PCR-positive animals among the wild boars found dead did not differ either (X^2^ (1, N = 368) = 0.9, *p* = 0.34). Only 2 (1.1%) of 244 re-examined seropositive animals but PCR-negative in the blood were found to be positive for the ASF virus genome in the bone marrow: a young male from Lithuania less than 12 months of age with a Ct value of 35 and a male from Latvia between one and two years old with a Ct value of 28. There was no significant difference between Lithuania and Latvia (X^2^ (1, N = 244) = 0.27, *p* = 0.61).

## 4. Discussion

Four categories of animals can be distinguished in a population based on their virological and serological status in the blood: (i) animals that are neither seropositive nor virus/genome-positive and are thus fully susceptible; (ii) animals that are only virus/genome-positive and therefore potential virus shedders that may infect other animals; (iii) animals that are virus/genome-positive and seropositive at the same time, and could excrete the virus and contribute to pathogen perpetuation; and finally, (iv) animals that are only seropositive.

The latter group of seropositive animals are suspected of being potential virus carriers, which can also excrete the virus and could thus play some role in the spread of the ASF virus. There are authors who support this hypothesis and others who see no epidemiological role of seropositive animals [5,6,7,8,9,10]. However, this is not a purely academic issue because it may have very practical consequences for disease control. A relevant question from the point of view of disease control is whether carriers, if they actually exist, are also capable of infecting other animals (being virus shedders) to such an extent that this is epidemiologically relevant, in the sense of maintaining and perpetuating the epidemic within a population and contributing to virus spread to uninfected populations. In some experimental studies, evidence was given that seropositive animals can occasionally infect other animals [6]. In other studies, this phenomenon could not be shown [7]. If seropositive animals actually have epidemiological relevance, then disease control measures would have to be tailored accordingly. If they are not relevant, control measures can target other transmission mechanisms, e.g., on viremic animals or on carcasses that should be removed from a population habitat even more efficiently.

The presented ratios of the proportions of seropositive and PCR-positive animals among hunted wild boars for Lithuania and Latvia for the period under consideration have also been described by other authors [18]. Most positive cases were found in the group of dead wild boars (Table 1), which is certainly an effect of targeted sampling of dead animals, most likely due to ASF. In total, 52% of the carcasses found in Latvia and 46% found in Lithuania were PCR-positive. This group of dead wild boars therefore poses a major threat in terms of virus spread. The 2 seropositive animals out of a total of 21,575 hunted animals that tested PCR-negative in the blood but positive in bone marrow are probably of very limited, if any, consequence, even if they would have been virus shedders. Furthermore, it is very doubtful whether these animals actually excreted virus during the convalescence period.

Interestingly, there are no significant differences between Lithuania and Latvia in the proportion of PCR-positive carcasses and the proportion of PCR-positive (bone marrow) and PCR-negative hunted wild boars. The latter could indicate common control and surveillance strategies in the infected areas of both countries.

The results obtained in this study clearly indicate that seropositive animals, which theoretically could also be virus shedders, are hardly found in the field and thus do not play an epidemiological role, at least not in the wild boar populations we studied. Their prevalence is too low to explain the maintenance of an epidemic on the scale that is occurring in these two countries.

Our results are also in line with the experimental studies in domestic pigs, where no convincing results for the epidemiological role of seropositive animals as virus shedders were found [11].

## 5. Conclusions

Considering the results of our study, we conclude that seropositive animals do not play a significant role from the point of view of disease control. Therefore, ASF control measures must be focused on viremic animals and carcasses, as well as on biosecurity measures during and after hunting.

## Figures and Tables

**Figure 1 pathogens-12-00723-f001:**
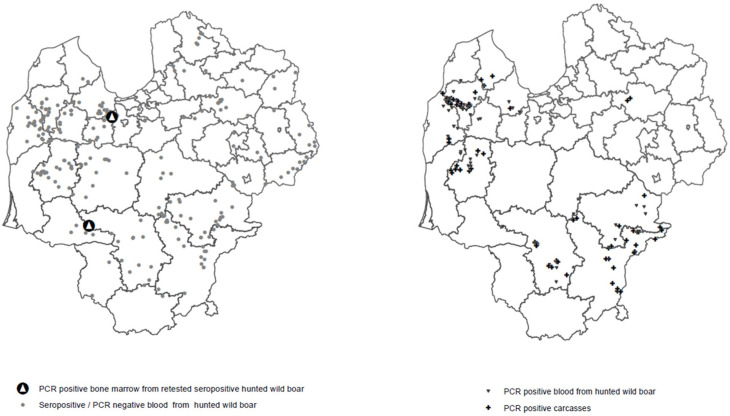
Overview of field study results and ASF cases in wild boars in Latvia and Lithuania in 2020.

**Table 1 pathogens-12-00723-t001:** Overview of virological and serological test results of ASF in wild boars tested in Latvia and Lithuania in 2020.

Country	Total Number of Blood Samples Tested from Hunted Wild Boars and Test Results (May–December 2020)					Re-Tested Hunted Animals ** (Bone Marrow)		Wild Boar Carcasses	
	n	Seronegative and PCR-Negative	Seronegative and PCR-Positive	** Seropositive and PCR-Negative	Seropositive and PCR-Positive	n	PCR-Positive	n	PCR-Positive
**Lithuania**	8020	7911 * 98.6, CI 98.3–98.9	1 0.01, CI 0.0–0.07	105 1.3, CI 1.1–1.6	3 0.04, CI 0.01–0.11	105	1 0.95, CI 0.0–5.2	177	82 46.3, CI 38.7–54.0
**Latvia**	13,555	13,363 98.6, CI 98.3–98.8	10 0.07, CI 0.04–0.14	139 1.0, CI 0.9–1.2	21 0.15, CI 0.10–0.24	139	1 0.72, CI 0.0–4.0	191	99 51.8, CI 44.4–59.1
**Total**	**21,575**	**21,274** **98.6, CI 98.4–98.8**	**11** **0.05, CI 0.03–0.09**	**244** **1.1, CI 1.0–1.3**	**24** **0.11, CI 0.07–0.17**	**244**	**2** **0.82, CI 0.0–3.0**	**368**	**181** **49.2, CI 43.9–54.5**

* Absolute number followed by proportion (%) with confidence interval (CI, 95%). ** Corresponding samples—Seropositive but PCR-Negative in blood samples.

## Data Availability

The original data used for this study can be obtained from the corresponding author after approval by the competent institutions in Latvia and Lithuania.

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
