# Peer review of "Do Seropositive Wild Boars Pose a Risk for the Spread of African Swine Fever? Analysis of Field Data from Latvia and Lithuania"

_pathogens, 2023, doi:10.3390/pathogens12050723_

Round 1

Reviewer 1 Report

This paper evaluates using samples material collected in 2020 during hunting season in Latvia and Lithuania, if seropositive wild boar poses a risk for the spread of African Swine fever. The authors show the results of PCR analysis performed on the bone marrow obtained from ASF seropositive animals as an indicator of virus persistence. According the discussions reported in the paper, the interpretation of the results of this study is limited because the specimens are coming from dead animals: only a designed study on live animals sampled twice with an approximately six-month interval and tested for the presence of ASF antibodies and ASFV genome can produce scientific data to confirm or denied the role of the wild board as long-term ASF-carriers especially in the field of the diseases control. A virus isolation test performed on the 2 bone marrow tested positive by PCR would have provided important data and given more support to the discussions and more impact to the publication.

The following points should be considered:

1) 
Materials and methods: please describe briefly the technique used to collect the bone marrow from the homers, the quantity of bone marrow used for the extraction of the DNA and the kit used for the extraction of the genetic material.

2) Line 109, Figure 1: correct “20204” in “2020”.

3) Line 114: In the text correct “virus positive” with “virus/genome positive”.

Author Response

Dear Reviewer,

Thank you for the positive feedback on our manuscript.

The following points should be considered:

1) Materials and methods: please describe briefly the technique used to collect the bone marrow from the homers, the quantity of bone marrow used for the extraction of the DNA and the kit used for the extraction of the genetic material.  – Description provided in the manuscript (lines 91-98)

2) Line 109, Figure 1: correct “20204” in “2020”. – Corrected  

3) Line 114: In the text correct “virus positive” with “virus/genome positive”. – Corrected (line 147).  

Reviewer 2 Report

Dear Author/s

Thank you for the research that you conducted, it will add to the body of knowledge on the subject, especially regarding regulatory matters.

I did however encounter difficulties in reading Table 1 because of the symbols that was used e.g  the dots (.) to separate hundreds and thousands, and the (/) in headings. It seemed like you were reporting fractions or decimals. I suggest that you mention in the first column, Total number tested and remove the dots. Also, for each column, remove the symbol that makes it look like you are reporting fractions and just use the word "and", or symbol "&".

My questions, comments and suggestions are as follows:

1. Introduction, Line 68: add "off" behind "disposed" so that it can read: "disposed off:

2. Remove the aim of the study from Materials and Methods and mention it at the end of the Introduction section.

3. Mention the dead boars found under materials and methods. The reader only finds out about them in the results but they were never mentioned before. Which samples were tested using PCR? If it was blood, was the plasma tested for antibodies? Were post mortems done and were there pathological signs consistent with ASF? Were they tested for other causes of death? If these were not done, state why and mention these as the short comings of the study. What were the ASF PCR Ct values.

4. Were the carcasses of the hunted boars examined for signs of disease and suitability for consumption i.e meat inspection. This is very important since there were PCR positives and deaths in the two study areas. If not, mention why.

5. Your paper lacks important epidemiological information. Were there active outbreaks of ASF in the two places during the sampling period (May to December)? Why was the time chosen? Active hunting season or height of outbreak , maybe in domestic pigs? 

6. Since you recorded the information, what were the percentages of male, female, young, old etc you tested? Both huunted and those which died (carcasses)

7. The paper lacks statistical analysis. These are needed to support the conclusion that you are making. You must do statistical analysis to show the readers associations/correlations/significance levels etc between the hunted boars and the dead carcasses, and comment on the short comings. Statistics must appear under materials and methods.

Regards,

The English is fine and readable.

Author Response

Dear Reviewer,

Thank you very much for the positive evaluation of our manuscript.

I did however encounter difficulties in reading Table 1 because of the symbols that was used e.g  the dots (.) to separate hundreds and thousands, and the (/) in headings. It seemed like you were reporting fractions or decimals. I suggest that you mention in the first column, Total number tested and remove the dots. Also, for each column, remove the symbol that makes it look like you are reporting fractions and just use the word "and", or symbol "&". – Suggested changes are made in Table 1.  

My questions, comments and suggestions are as follows: 

1. Introduction, Line 68: add "off" behind "disposed" so that it can read: "disposed off:  - Correction made (line 70) 

2. Remove the aim of the study from Materials and Methods and mention it at the end of the Introduction section. – Corrected (lines 79-80)

3. Mention the dead boars found under materials and methods. The reader only finds out about them in the results but they were never mentioned before. Which samples were tested using PCR? If it was blood, was the plasma tested for antibodies? Addressed under Materials and methods section (lines 112-116). Were post mortems done and were there pathological signs consistent with ASF? Were they tested for other causes of death? If these were not done, state why and mention these as the short comings of the study. Post mortems were done only for those carcasses that were fresh enough. In most cases carcasses of dead wild boar were decomposed and only samples available were long bones (bone marrow were used for laboratory testing by PCR). Classical swine fever was the only differential diagnosis that were tested by PCR, however it was used only in carcasses where ASF result was negative. What were the ASF PCR Ct values. – Samples with Ct values <40 were considered as PCR positive for ASF. 

4. Were the carcasses of the hunted boars examined for signs of disease and suitability for consumption i.e meat inspection. This is very important since there were PCR positives and deaths in the two study areas. If not, mention why. – Most of wild boar hunted in Latvia and Lithuania are used for private consumption by hunters – so there is no official meet inspection. Only very limited number of hunted wild boar are entering the food chain and they are passing official meat inspection. It is important to mention, that all hunted wild boar must be sampled and tested to ASF virus genome and antibodies and only in case of both tests are negative – meet can be consumed. If one of the laboratory test results is positive, carcass is disposed off. 

5. Your paper lacks important epidemiological information. Were there active outbreaks of ASF in the two places during the sampling period (May to December)? The location of active ASF outbreaks in hunted and found dead wild boar during 2020 are presented in Figure 1.  Why was the time chosen? Active hunting season or height of outbreak, maybe in domestic pigs? - The beginning of sampling period was chosen mainly due to management reasons – when the study was prepared, and people involved were trained. The end of the sampling period was the end of the implementation of routine ASF surveillance program for 2020.

6. Since you recorded the information, what were the percentages of male, female, young, old etc you tested? Both hunted and those which died (carcasses) – Data on animal sex and age group were collected as a standard information for each wild boar sampled. Initially, we though it would be good dataset for analysis of PCR positive results (in bone marrow) but since there were only two positive results we presented sex, age and Ct value for each animal in the text (lines 131-135). Therefore, we agree that mentioning sex and estimated age of wild boar under Materials and methods without further analysis is irrelevant and we excluded them (line 101). 

The analysis of percentages of male, female, age group of hunted and found dead wild boar tested was not the aim of this study. This kind of analysis are described in our previous studies e.g., 

-        Schulz, K, Oļševskis, E, Viltrop, A, et al. Eight Years of African Swine Fever in the Baltic States: Epidemiological Reflections. Pathogens. 2022;11(6):711. Published 2022 Jun 20. doi:10.3390/pathogens11060711  

7. The paper lacks statistical analysis. These are needed to support the conclusion that you are making. You must do statistical analysis to show the readers associations/correlations/significance levels etc between the hunted boars and the dead carcasses, and comment on the short comings. Statistics must appear under materials and methods. – Methods used for statistical analysis are now described under the section of Materials and methods (lines 116-120).  The results of statistical analysis are described under Results section (lines 122-136). Confidence intervals (CI, 95%) are included in Table 1. The results of statistical analysis are discussed in Discussion section (lines 169-183). 

Round 2

Reviewer 1 Report

The authors added the missing information and made all the corrections. The article is eligible for publication.